# Efficacy of Inactivation of Human Enteroviruses by Dual-Wavelength Germicidal Ultraviolet (UV-C) Light Emitting Diodes (LEDs)

**Hyoungmin Woo [1], Sara E. Beck [2,†], Laura A. Boczek [1], Kelsie M. Carlson [1,‡], Nichole E. Brinkman [1], Karl G. Linden [2] , Oliver R. Lawal [3], Samuel L. Hayes [1] and Hodon Ryu [1,\*]**

[1] United States Environmental Protection Agency, Office of Research and Development, 26 W. Martin Luther King Dr., Cincinnati, OH 45268, USA; hyoungmin.woo@gmail.com (H.W.); boczek.laura@epa.gov (L.A.B.); Kelsie.Carlson@weasengineering.com (K.M.C.); Brinkman.Nichole@epa.gov (N.E.B.); shayes936@gmail.com (S.L.H.)

[2] Department of Civil, Environmental, and Architectural Engineering, University of Colorado Boulder, UCB 428, Boulder, CO 80309, USA; sara.beck@colorado.edu (S.E.B.); karl.linden@colorado.edu (K.G.L.)

[3] AquiSense Technologies, 4400 Olympic Boulevard, Erlanger, KY 41018, USA; oliver.lawal@aquisense.com

\* Correspondence: ryu.hodon@epa.gov; Tel.: +1-513-487-2062

† Present address: Department of Environmental Microbiology, Eawag: Swiss Federal Institute of Aquatic Science and Technology, 8600 Dubendorf, Switzerland.

‡ Present address: Weas Engineering, Inc., 17297 Oak Ridge Rd, Westfield, IN 46074, USA.

**Abstract:** The efficacy of germicidal ultraviolet (UV-C) light emitting diodes (LEDs) was evaluated for inactivating human enteroviruses included on the United States Environmental Protection Agency (EPA)'s Contaminant Candidate List (CCL). A UV-C LED device, emitting at peaks of 260 nm and 280 nm and the combination of 260/280 nm together, was used to measure and compare potential synergistic effects of dual wavelengths for disinfecting viral organisms. The 260 nm LED proved to be the most effective at inactivating the CCL enteroviruses tested. To obtain 2-$\log_{10}$ inactivation credit for the 260 nm LED, the fluences (UV doses) required are approximately 8 mJ/cm$^2$ for coxsackievirus A10 and poliovirus 1, 10 mJ/cm$^2$ for enterovirus 70, and 13 mJ/cm$^2$ for echovirus 30. No synergistic effect was detected when evaluating the log inactivation of enteroviruses irradiated by the dual-wavelength UV-C LEDs.

**Keywords:** ultraviolet disinfection; dual-wavelength; UV-C LEDs; human enteroviruses; viral inactivation efficacy; synergy

## 1. Introduction

Human enteroviruses are a significant cause of waterborne disease, resulting in gastrointestinal and upper respiratory tract infections, as well as more severe illnesses, such as viral meningitis and encephalitis [1,2]. Non-polio enteroviruses cause about 10 to 15 million infections each year in the United States [3]. Enterovirus 71, as well as poliovirus, were listed in the top five global infectious disease threats determined by the Centers for Disease Control and Prevention (CDC) [4]. While most infected people with non-polio enteroviruses have mild illness, these viruses can cause infections in infants and other immunocompromised individuals with serious complications [3]. Human enteroviruses are often detected in wastewater effluents [5–7]. Since 2003, these viruses have been listed on the United States Environmental Protection Agency (USEPA)'s Contaminant Candidate

List (CCL) as waterborne pathogens that could warrant inclusion in future regulations under the Safe Drinking Water Act [8].

Drinking water treatment plants have the disinfection capabilities to achieve effective viral inactivation through various disinfection barriers during the course of treatment processes. Unlike chemical disinfectants, such as chlorine and ozone, ultraviolet (UV) light has been successfully adapted for treating these waterborne pathogens without the formation of carcinogenic disinfection by-products (DBPs) in drinking water treatment systems [9–11]. However, conventional low-pressure (LP) mercury vapor lamps have some practical limitations in water treatment applications, such as limitations in energy efficiency and, more importantly, potential mercury contamination from accidental breakage or improper disposal of the lamps.

Emerging UV light emitting diodes (LEDs) technology has enormous potential in potable water disinfection applications. UV LEDs are well suited for point-of-use (POU) devices since LEDs are smaller, lighter, less fragile, and mercury-free [12]. Moreover, UV LEDs offer the flexibility to use preferred germicidal wavelengths, which range from 254–280 nm (i.e., germicidal ultraviolet (UV-C)) [13–19]. Extensive studies have been conducted on microorganism inactivation using UV-C LEDs [20–25]. However, most studies have focused primarily on microbial indicators and, to date, limited studies targeting waterborne pathogens have been reported. Most recently, Rattanakul and Oguma [25] showed inactivation efficacy of *Pseudomonas aeruginosa* and *Legionella pneumophila*, using UV LEDs with peak wavelength emissions at 265 and 280 nm. Beck et al. [26] reported comprehensive results of dual-wavelength UV-C LEDs emitting at peaks of 260 nm, 280 nm, and the combination of 260/280 nm together against a suite of waterborne microorganisms, including human adenovirus, which is one of the most resistant pathogens to UV irradiation. To our knowledge, the efficacy of inactivation of other human enteric viruses by a polychromatic light spectrum of UV-C LED has not yet been reported. The main objective of this study was to investigate the efficacy of dual-wavelength UV-C LEDs for inactivating four serotype representatives of human enterovirus species.

## 2. Materials and Methods

Representative serotypes of the four human enteric species (*Enterovirus* A–D) [27] were selected as test viruses, including coxsackievirus A10 (CVA10, Kowalik strain), echovirus 30 (Echo30, Bastianni strain), poliovirus 1 (PV1, Mahoney strain), and enterovirus 70 (EV70, J670/71 strain) respectively. The enteroviruses were obtained from the American Type Culture Collection (Manassas, VA, USA) and propagated in buffalo green monkey kidney (BGMK) cells, as described previously [28]. UV-exposure experiments against these enteroviruses were performed as described previously [26]. Briefly, bench-scale performance evaluation was conducted using a collimated beam (CB) apparatus with LEDs with peak emissions of 260 nm, 280 nm, and the combination of 260/280 nm together. The incident irradiances of the CB apparatus at the center of the dish, measured at the weighted average wavelengths of the three emissions with an ILT1400 radiometer and SED240/W detector (International Light Technologies, Peabody, MA, USA), were 0.194 mW/cm$^2$, 0.314 mW/cm$^2$, and 0.473 mW/cm$^2$ for the 260 nm LED, the 280 nm LED, and the combination of 260/280 nm together (38%—260 nm and 62%—280 nm), respectively. The applied average fluences throughout each water sample were determined according to published methods for CB tests with polychromatic sources [26,29,30]. Fluence calculations incorporated the incident irradiance, the UV LED emission spectra, divergence of the light, reflection off the surface of the water, non-uniformity of the light across the petri dish (petri factor of 0.87–0.93), and the water absorbance.

Triplicate CB tests were performed with mixed stocks of four viruses. Infectious virus concentrations were determined using an integrated cell culture reverse transcriptase quantitative polymerase chain reaction (ICC-RTqPCR), as described previously [31,32]. Log$_{10}$ inactivation of enteroviruses (I) is defined by Equation (1):

$$\mathrm{I} \; = \; -\log_{10}\!\left(\frac{N_d}{N_0}\right) \tag{1}$$

where $N_0$ and $N_d$ (MPN/mL) are the initial concentration and the concentration of infectious enteroviruses after a specified UV dose, respectively. The $\log_{10}$ inactivation was estimated for four human enteroviruses at four different UV fluences (5 mJ/cm$^2$, 10 mJ/cm$^2$, 15 mJ/cm$^2$, and 20 mJ/cm$^2$), using a dual-wavelength UV-C LED device emitting a polychromatic light spectrum with the peaks at 259.6 nm and 276.6 nm. For the comparison of UV dose response of enteroviruses to low-pressure (LP) UV at 254 nm, the $\log_{10}$ inactivation rates of LP UV were adapted from Ryu et al. [32].

Analysis of variance (ANOVA) tests were performed to determine if there was a significant difference in inactivation efficacy of the tested four enteroviruses among different wavelength spectra (e.g., 260 nm, 280 nm, and 260/280 nm). *P* values of <0.05 were considered statistically significant.

## 3. Results and Discussion

The CB experiments were designed to test the effectiveness of irradiation from individual LED 260 nm and 280 nm and the simultaneous irradiation of LED 260/280 nm at inactivating the selected viruses. The $\log_{10}$ inactivation results after each of the irradiation scenarios (e.g., 260 nm, 280 nm, and 260/280 nm) to each of the four viruses are presented in Figure 1. The tailing for Echo30 and EV70 was observed at 15 mJ/cm$^2$ of UV fluence, possibly due to assay limitations (i.e., detection limit). To achieve a 2-$\log_{10}$ reduction in infectious virus (i.e., 99% reduction) by the most effective wavelength, 260 nm LED, averaged UV doses were approximately 8 mJ/cm$^2$ for CVA10 and PV1, 10 mJ/cm$^2$ for EV70, and 13 mJ/cm$^2$ for Echo30 (Figure 1). In comparison, for a 2-$\log_{10}$ reduction using 280 nm LED, the required doses were averaged to approximately 12 mJ/cm$^2$ for CVA10 and EV70, 11 mJ/cm$^2$ for PV1, and 15 mJ/cm$^2$ for Echo30 (Figure 1), indicating that light at a wavelength of 280 nm was less effective for inactivating enteroviruses.

The 5 mJ/cm$^2$ of UV dose using 260 nm LED can provide at least 1-$\log_{10}$ inactivation of all the enteroviruses tested (Figure A1). When 280 nm LED was used for the same UV dose, only EV70 showed over 1-$\log_{10}$ inactivation, whereas less than 1-$\log_{10}$ inactivation of the other enteroviruses was achieved. EV70 at 280 nm light spectrum showed the best efficacy of $\log_{10}$ inactivation but significantly less inactivation efficacy than that of 260 nm irradiation (i.e., 1.1 vs. 1.6 $\log_{10}$ reduction for 5 mJ/cm$^2$ of UV fluence, $p = 0.01$). At 280 nm light spectrum, the other viruses showed relatively low performance with $\log_{10}$ reduction range of 0.5–0.8 (Figure A1). Simultaneous irradiation at 260/280 nm LED was either as effective as 260 nm alone (EV70, Echo30) or less effective (CVA10, PV1). In addition, all UV-C LED wavelengths were more effective than LP-UV at 254 nm (Figure 1), which supports previously published reports [32,33]. Gerba et al. [33] showed that a UV dose of 14 mJ/cm$^2$ to 18 mJ/cm$^2$ resulted in a 2-$\log_{10}$ inactivation credit of enteric viruses, excluding adenovirus using LP mercury vapor UV lamp. These UV doses for enteric viruses tested yielded much greater inactivation rate constants than viral indicators (e.g., MS2 and Qβ bacteriophages) [25,26], suggesting that these bacteriophages could be used as conservative viral indicators in UV disinfection studies.

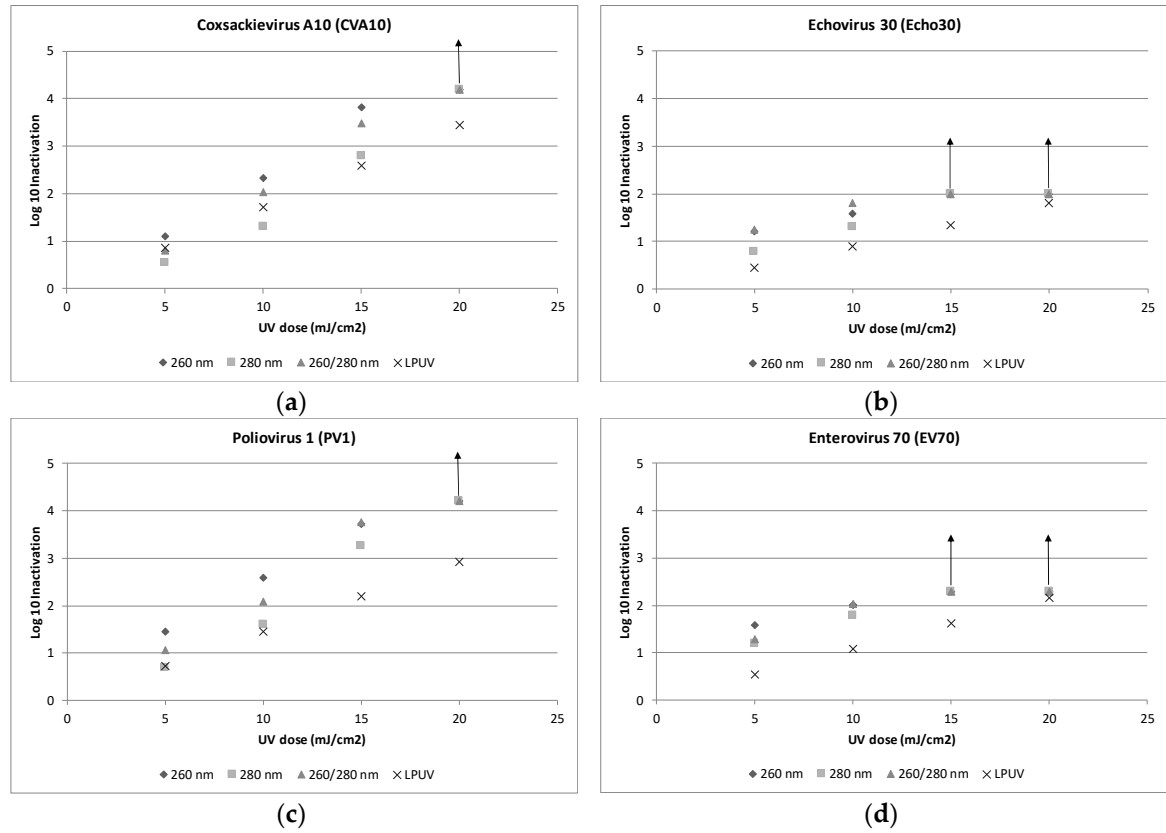

**Figure 1.** Log$_{10}$ inactivation of four human enteroviruses after exposure to germicidal ultraviolet light emitting diodes (UV-C LEDs) and low-pressure UV (LPUV) at 254 nm ((**a**) Coxsackievirus A10, (**b**) Echovirus 30, (**c**) Poliovirus 1, and (**d**) Enterovirus 70). Arrow (↑) represents detection limit. Each data point is an arithmetic average of log$_{10}$ inactivation from independent triplicate collimated beam (CB) tests. Refer to Figure A1 for more detailed statistics, including standard deviation. The log$_{10}$ inactivation rates LPUV at 254 nm were adapted from Ryu et al. [32].

Overall, the 260 nm light spectrum was most effective at inactivating all the enteroviruses tested, followed by the 260/280 nm light spectrum, and lastly, the 280 nm light spectrum (Figure 1). These results support our previous study with MS2 bacteriophage (an RNA virus), which also reported no synergistic inactivation of RNA viruses by the 260/280 nm combination [26]. Most recently, Rattanakul and Oguma [25] reported that 265 nm UV-LED was the most effective fluence for disinfecting bacterial pathogens and Qβ bacteriophage (an RNA virus) when compared to 280 and 300 nm UV-LEDs and LP UV at 254 nm. Several studies have also shown that the MS2 virus is more susceptible to UV light at 260 nm than at 280 nm [34,35]. A relative peak at 260 nm for the UV absorbance of MS2 RNA and in the MS2 action spectrum [34] indicates that this wavelength is most effective for viral RNA damage. The sufficient fluence of UV light at specific nucleic acid absorbing wavelengths inactivates microorganisms by impeding the replication of their DNA or RNA molecules [9,36,37]. While nucleic acids absorb UV light between 240 and 280 nm, both DNA and RNA have peak adsorption at or near 260 nm [20]. Unlike conventional monochromatic UV light from a LP mercury vapor lamp (at 254 nm), polychromatic UV-C LED produces a broader band of light emission. For example, an LED with a 260-nm peak emission wavelength has a spectral range of 250 nm to 270 nm, which covers the peak nucleic acid adsorption range of nucleic acid molecules. However, peak absorption distribution is dependent on the specific target organism that has an absorption maximum between 254 and 280 nm [9,18]. On the other hand, human adenovirus (a DNA virus) showed relatively high inactivation efficacy at 280 nm [26]. Given that the UV absorbance of protein has a relative peak near 280 nm [38], protein damage plays an important role in adenovirus inactivation [39]. Unlike

human adenovirus, human enteroviruses showed less resistance to UV light of 280 nm, suggesting a less important role of viral proteins in the infectious process. Further study on viral inactivation mechanisms across the germicidal UV spectrum is needed.

## 4. Conclusions

This research utilized a germicidal UV-C LED device emitting a polychromatic light spectrum around the peak at 260 and 280 nm to evaluate its efficacy at inactivating human enteroviruses in water. The comparison of $\log_{10}$ inactivation of microorganisms irradiated individually by 260 and 280 nm UV LED units and the $\log_{10}$ inactivation achieved from the combined 260/280 irradiation shows no synergistic effects. Irradiation of 260 nm peak light spectrum is more effective for the inactivation of human enteroviruses. Overall, UV LEDs showed the capability to effectively inactivate the CCL enteroviruses tested. The higher efficacy of the 260 nm LED encourages further studies on its applicability for sustainable water treatment and other CCL pathogens. UV-LEDs also have several practical advantages that allow for a device effective at the point of use (POU). This would disinfect drinking water prior to public consumption. As waterborne pathogens continue to pose a public health threat, the development of novel technologies—such as LED POU devices—will be important to promote safe drinking water.

**Author Contributions:** Conceptualization, K.G.L., O.R.L., S.L.H., and H.R.; Methodology, H.W., S.E.B., L.A.B., K.M.C., and N.E.B.; Validation, H.W. and S.E.B.; Formal Analysis, H.W. and S.E.B.; Investigation, H.W., S.E.B., L.A.B., K.M.C., and N.E.B.; Resources, K.G.L., O.R.L., and H.R.; Data Curation, H.W. and S.E.B.; Writing—Original Draft Preparation, H.W.; Writing—Review & Editing, H.R.; Supervision, K.G.L. and H.R.; Project Administration, S.L.H. and H.R.

**Funding:** This research received no external funding.

**Acknowledgments:** We thank Jeongwon Ryu for an editorial review of this manuscript. The U.S. Environmental Protection Agency, through its Office of Research and Development, funded and managed the research described herein. This work has been subjected to the agency's administrative review and has been approved for external publication. Any opinions expressed in this paper are those of the authors and do not necessarily reflect the views of the agency; therefore, no official endorsement should be inferred. Any mention of trade names or commercial products does not constitute endorsement or recommendation for use.

**Conflicts of Interest:** The authors declare no conflict of interest.

## Appendix A

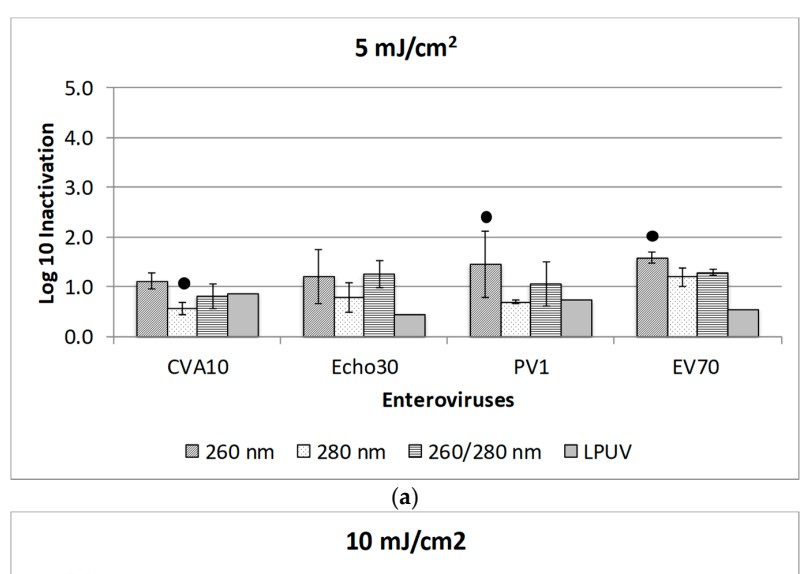

(a)

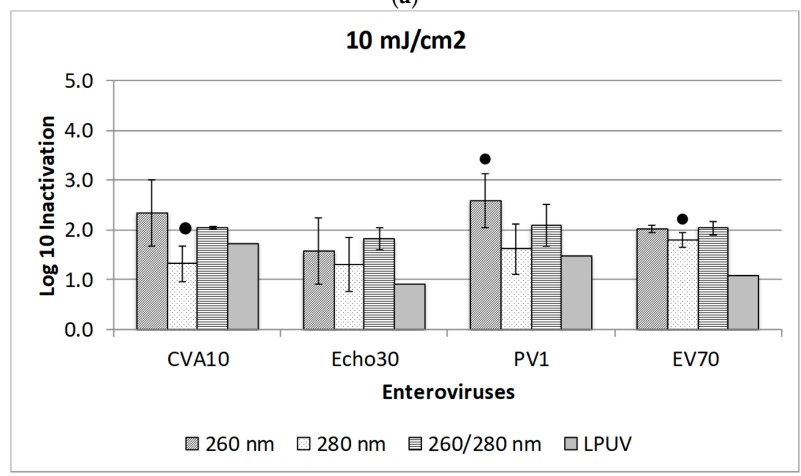

(b)

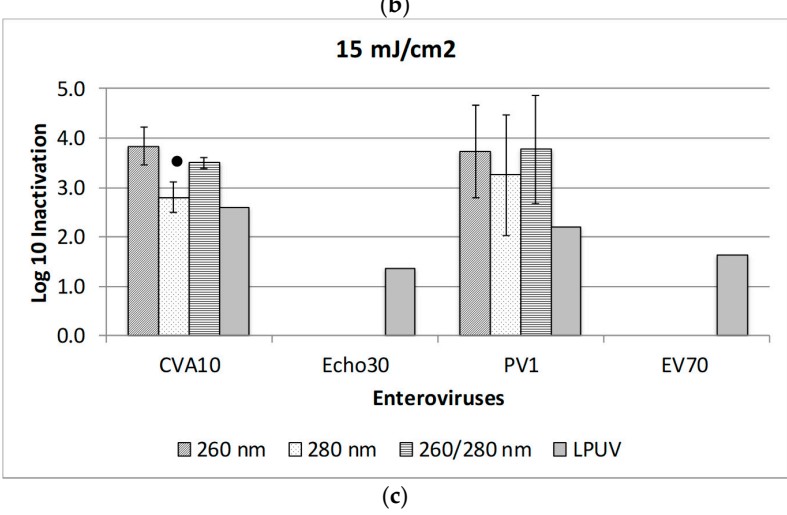

(c)

**Figure A1.** Comparison results of $\log_{10}$ inactivation rate from the 5 mJ/cm$^2$ (**a**), 10 mJ/cm$^2$ (**b**), and 15 mJ/cm$^2$ (**c**) irradiation of UV-C LEDs for four human enteroviruses. The error bars represent 1 standard deviation. Symbol (●) represents the *p* value of <0.05 among three wavelengths, as determined by ANOVA. Echo30 and EV70 for a UV dose of 15 mJ/cm$^2$ were not determined. The $\log_{10}$ inactivation rates by low-pressure UV (LPUV) at 254 nm were estimated using UV dose–response curves with a UV dose range of 10–30 mJ/cm$^2$, adapted from Ryu et al. [32].

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
