# Peer review of "Efficacy of Inactivation of Human Enteroviruses by Dual-Wavelength Germicidal Ultraviolet (UV-C) Light Emitting Diodes (LEDs)"

_water, doi:10.3390/w11061131_

Round 1
Reviewer 1 Report
This is very good paper that may be published "as is" after correcting the obvious small errors like the missing asterisk symbol that is supposed to be in the parenthesis in the caption of Fig. A1 and 'an LED' replaced by 'a LED' in L143.
However, in order to increase impact, I would suggest that the two following issues may be discussed in more detail:
How "clean" must the water be for this technology to work in real life? Eg. what levels of organic (and inorganic?) matter, of (other) viruses and bacteria? Potable water is mentioned early, but this may be a broad category and not specific.
How easily is this implemented for industrial applications? Which applications do you foresee? Can this also be used in Recirculating Aquaculture Systems or for cleaning of wastewater or processing water?
Author Response
This is very good paper that may be published "as is" after correcting the obvious small errors like the missing asterisk symbol that is supposed to be in the parenthesis in the caption of Fig. A1 and 'an LED' replaced by 'a LED' in L143.
Our response: We appreciate the reviewer for her/his positive remarks and comments. Changed as suggested.
However, in order to increase impact, I would suggest that the two following issues may be discussed in more detail:
How "clean" must the water be for this technology to work in real life? Eg. what levels of organic (and inorganic?) matter, of (other) viruses and bacteria? Potable water is mentioned early, but this may be a broad category and not specific.
How easily is this implemented for industrial applications? Which applications do you foresee? Can this also be used in Recirculating Aquaculture Systems or for cleaning of wastewater or processing water?
Our response: We thank the reviewer for her/his suggestions. Indeed, conventional UV technology has been used in wastewater treatment utilities for several decades, even longer than drinking water industries. As stated in the text, although emerging UV LED technology has a lot of potential in water disinfection applications, many issues including 1) scaling up and 2) its microbial inactivation efficacy in different water matrices still remain to be tacked for its universal use as a water treatment process. In this study, we focused on POU application for drinking water. As suggested, we have included the following discussion at the end of Conclusion section: “UV-LEDs also have several practical advantages which allow for a device effective at the point-of-use (POU). This would disinfect drinking water prior to public consumption. As waterborne pathogens continue to pose a public health threat, the development of novel technologies – such as LED POU devices – will be important to promote safe drinking water.”
Reviewer 2 Report
Error bar seemed to be missing in Figure 1 and improved the quality of Figure 1 would be good.
Author Response
Error bar seemed to be missing in Figure 1 and improved the quality of Figure 1 would be good.
Our response: We agree that error bars represent statistically sound data sets. In Figure 1, we attempted to show comparative inactivation rates of different wavelengths against all the testing enteroviruses. We thought the addition of error bars in the figure makes it more complex. However, we absolutely understand the reviewer’s concern, and instead please note that error bars for all the tested UV doses except for 20 mJ/cm2 were presented in Figure A1.
Reviewer 3 Report
Review of Manuscript No. water-503839
Title: “Efficacy of inactivation of human enteroviruses by dual-wavelength germicidal ultraviolet (UV-C) light emitting diodes (LEDs)”
Authors: Hyoungmin Woo, et al.
The authors studied important topic related to the inactivation of human enteroviruses versus ultraviolet light emitting diodes. However, following points should be revised prior to publication.
(1) During the review of the above submitted paper, we found very similar abstract by the same authors listed below but in the different paper.
Title: Efficacy of Inactivation of Human Enteroviruses by Multiple-Wavelength UV LEDs - abstract.
https://cfpub.epa.gov/si/si_public_record_report.cfm?dirEntryId=319574&Lab=NRMRL.
However, this paper is not included in the reference of the submitted paper.
The authors should clearly describe the difference between the previously published paper and the present submitted paper by refereeing the previously published paper in order to avoid the duplication.
(2) To quantitatively determine the synergy effect, 2-Log data is insufficient, and at least 3- to 5-Log data is necessary. (Many studies show higher inactivation data than 3-Log.)
(3) The intensity ratio of combined LED is not clear. For example, at the total UV dose of 10 mJ/cm2, do you use 50 %-260nm and 50 %-280 nm UV dose? The portion of UV dose for each wavelength is not clear.
(4) To quantitatively determine the synergy effect, error bars for all data points of Figure 1 are necessary.
(5) To quantitatively determine the synergy effect, quantitative description of P values are required.
(6) The efficacy of synergy effect is not precisely discussed based on other references.
This paper is not appropriate for a Communication in Water at the present stage. Major revisions that solves above points are necessary prior to publication.

Author Response
The authors studied important topic related to the inactivation of human enteroviruses versus ultraviolet light emitting diodes. However, following points should be revised prior to publication.
Our response: We thank the reviewer for her/his comments.
(1) During the review of the above submitted paper, we found very similar abstract by the same authors listed below but in the different paper.
Title: Efficacy of Inactivation of Human Enteroviruses by Multiple-Wavelength UV LEDs - abstract.
https://cfpub.epa.gov/si/si_public_record_report.cfm?dirEntryId=319574&Lab=NRMRL.
However, this paper is not included in the reference of the submitted paper.
The authors should clearly describe the difference between the previously published paper and the present submitted paper by refereeing the previously published paper in order to avoid the duplication.
Our response: As implied by the reviewer, some of the results were presented in a conference abstract. We would like to note that we presented preliminary data at the 2016 ASM general meeting (poster presentation) in Boston, MA USA. This manuscript represents the completed research work.
(2) To quantitatively determine the synergy effect, 2-Log data is insufficient, and at least 3- to 5-Log data is necessary. (Many studies show higher inactivation data than 3-Log.)
Our response: The reviewer brings up a good point. We agree that many studies have shown >3-log inactivation. For this study, we originally designed the experiments to achieve >3-log inactivation of all four enteroviruses. As stated in the text, whereas log10 inactivations of coxsackievirus and poliovirus exceed 3-log at 15 mJ/cm2 of UV fluence, the tailing for the other echovirus and enterovirus was observed at 15 mJ/cm2, possibly due to assay limitations. In the future work, appropriate spiking levels of microbes should be considered, which makes the research much stronger.
(3) The intensity ratio of combined LED is not clear. For example, at the total UV dose of 10 mJ/cm2, do you use 50 %-260nm and 50 %-280 nm UV dose? The portion of UV dose for each wavelength is not clear.
Our response: As suggested, we have included relevant information in the section of Materials and Methods. Briefly, the ratio of 260 nm and 280 nm can be calculated by both measured irradiances, resulting in 0.62 (=0.194/0.314) which corresponds 38%-260 nm and 62%-280 nm.
(4) To quantitatively determine the synergy effect, error bars for all data points of Figure 1 are necessary.
Our response: We thank the reviewer for her/his comments. Reviewer 2 pointed out the same issue. As stated in our response to Reviewer 2’s comment, we agree that error bars represent statistically sound data sets. In Figure 1, we attempted to show comparative inactivation rates of different wavelengths against all the testing enteroviruses. We thought the addition of error bars in the figure makes it more complex. Rather, error bars for all the tested UV doses except for 20 mJ/cm2 were presented in Figure A1.
(5) To quantitatively determine the synergy effect, quantitative description of P values are required.
Our response: As the reviewer points out, P values are required for determining whether a result is statistically significant. As suggested, we have added absolute P values in the text.
(6) The efficacy of synergy effect is not precisely discussed based on other references.
Our response: We are not sure we understand this comment properly, but we have tried our best to address it as follows. As discussed in the text, 260 nm LED showed the best efficacy of log10 inactivation against all the tested viruses compared to 280 nm LED and the combined 260|280 LEDs. Moreover, unlike our previous UV-LED study (Beck et al., 2017 [26]) and many studies by other researchers, we had relatively limited results which make more rigorous discussion difficult. This is one of the biggest reasons why this manuscript was submitted as short Communication. In this study, we simply presented the inactivation efficacy with limited discussion of synergy effect, but we still believe this manuscript presents novel observations (the first UV-LED study targeting human enteroviruses) which would be beneficial to readers.
Round 2
Reviewer 3 Report
The authors studied important topic related to the inactivation of human enteroviruses versus ultraviolet light emitting diodes.
This manuscript might be accepted in the present form.
Comments)
However, if you think of general readers, your replies to following comments will be very helpful.
(1) The intensity ratio of two LEDs takes special value (38%-260 nm and 62%-280 nm), and it is helpful to describe the reason why you choose this ratio in order to give clearly understanding to general readers.
(2) The size of the characters in Figure 1 is small and it is difficult to find them out.